

# A computational framework for colour metrics and colour space transforms

Ivar Farup

Faculty of Computer Science and Media Technology, Gjøvik University College, Norway

## ABSTRACT

An object-oriented computational framework for the transformation of colour data and colour metric tensors is presented. The main idea of the design is to represent the transforms between spaces as compositions of objects from a class hierarchy providing the methods for both the transforms themselves and the corresponding Jacobian matrices. In this way, new colour spaces can be implemented on the fly by transforming from any existing colour space, and colour data in various formats as well as colour metric tensors and colour difference data can easily be transformed between the colour spaces. This reduces what normally requires several days of coding to a few lines of code without introducing a significant computational overhead. The framework is implemented in the Python programming language.

## INTRODUCTION

Colour data such as measured colours, specified colours or pixels of colour images are most commonly described as sets of points in a three-dimensional space—a so-called colour space. Many different colour spaces are currently in use in various settings. For many applications, selecting the best colour space for processing the data can be crucial (*Plataniotis & Venetsanopoulos, 2000*). Converting between all the different colour spaces can be challenging. Different conventions for scaling and normalisation is used, and many of the colour spaces commonly in use are inaccurately defined. The complexity of conversion is particularly present for computations involving colour metric data, which, by nature, is tensorial (*Deza & Deza, 2009*), giving rise to the need for not only the direct transformations, but also the corresponding Jacobian matrices—a tedious and error-prone process (*Pant & Farup, 2012*). So far, no common framework for such transformations of colour data and metrics including the automated computation of Jacobian matrices has been constructed.

From other fields of computational science, it is well established that object-oriented frameworks can be useful for simplifying such matters (*Beall & Shephard, 1999*). With the advent of modern high-level interpreted languages, the computational overhead is not nearly as high as before, and the ease of use has increased significantly (*Cai, Langtangen & Moe, 2005*). Thus, in order to simplify the matters for colour science and engineering, an object-oriented framework for colour space construction, and conversion of colour

Corresponding author
Ivar Farup, ivar.farup@ntnu.no

data and colour metric tensor data is designed. The framework is currently limited to three-dimensional colour spaces.

Following the background material on the principles of transforming colour data and related tensorial data in the following section, the principles and ideas underlying the framework are presented. To demonstrate to which degree the framework simplifies the implementation of colour data and metric transformations, an implementation of the framework using the high-level programming language Python (*Van Rossum & Drake, 1995*) is applied to some standard example problems.

## BACKGROUND

### Transformation of colour data

Transformations between different colour spaces can in general take the shape of a function, $\bar{x} = \bar{x}(x)$, where $x = (x_1, x_2, x_3)^T$ represents a colour, i.e., a point in a colour space. Fortunately, most common colour space conversions are made up of a small set of relatively simple mathematical operations. The linear transformation is a very common ingredient in the transforms. Some colour spaces, such as, e.g., the CIECAT02 colour adaptation space (*Moroney et al., 2002*), are even defined simply by a linear transformation from some other colour space:

$$\bar{x} = Ax, \tag{1}$$

where $A$ is a $3 \times 3$ constant matrix. Combined with the so-called gamma correction, which is applied channel-wise, most RGB type colour spaces, and also, e.g., the IPT (*Ebner & Fairchild, 1998*) colour space can be construced

$$\bar{x}_i = \mathrm{sgn}(x_i)|x_i|^\gamma, \tag{2}$$

where $\gamma > 0$ is the constant exponent.

For many perceptual colour spaces such as CIELAB, both Cartesian and cylindrical coordinates are commonly used for describing the chromatic plane. The transformation from Cartesian to polar is

$$\bar{x}_1 = x_1,$$
$$\bar{x}_2 = \sqrt{x_2^2 + x_3^2}, \tag{3}$$
$$\bar{x}_3 = \mathrm{atan2}(x_3, x_2),$$

with the corresponding inverse transform

$$\bar{x}_1 = x_1,$$
$$\bar{x}_2 = x_2 \cos(x_3), \tag{4}$$
$$\bar{x}_3 = x_2 \sin(x_3).$$

Chromaticities and luminances are often represented in projective spaces such as xyY,

$$\bar{x}_1 = \frac{x_1}{x_1 + x_2 + x_3},$$
$$\bar{x}_2 = \frac{x_2}{x_1 + x_2 + x_3}, \tag{5}$$
$$\bar{x}_3 = x_2.$$

Colour spaces used for colour metrics such as $\Delta E_E$ (*Oleari, Melgosa & Huertas, 2009*) and the various DIN99 metrics (*Cui et al., 2002*) often include a logarithmic compression of some or all of the channels such as lightness and chroma (radius in polar coordinates):

$$\bar{x}_i = a_i \ln(1 + b_i x_i), \tag{6}$$

where $a_i$ and $b_i$ are the parameters of the transform. Recently, the Poincaré disk representation of the hyperbolic plane has been used for representing the chromatic plane (*Lenz, Carmona & Meer, 2007*; *Farup, 2014*). The chroma-preserving mapping to the Poincaré disk can be written as a mapping of the radius in polar coordinates as

$$\bar{x}_2 = \tanh\left(\frac{x_2}{2R}\right), \tag{7}$$

where $R > 0$ is the radius of curvature. Besides these more generic transformations, various non-linear transformation functions specific to individual colour spaces are used in such cases as sRGB, CIELAB, CIELUV, the underlying colour space of the CIEDE2000 metric, etc.

## Transformation of tensorial data

Most colour metrics can be represented in the form of a line element, or a differential quadratic form (*Wyszecki & Stiles, 1982*, Chapter 8.4), as

$$ds^2 = dx^T G dx. \tag{8}$$

Here, G is the metric tensor—a function of the coordinates. For metrics defined as Euclidean distances in a given colour space, the metric tensor is the identity tensor, I, in the given space. Some colour metrics, like, e.g., CIEDE2000, cannot be written in this form, but can be linearised—or Riemannised—to a good approximation (*Pant & Farup, 2012*).

Under a coordinate transformation, $\bar{x} = \bar{x}(x)$, this metric transforms according to

$$ds^2 = dx^T G dx = d\bar{x}^T \frac{\partial x}{\partial \bar{x}}^T G \frac{\partial x}{\partial \bar{x}} d\bar{x} = d\bar{x}^T \bar{G} d\bar{x}, \tag{9}$$

where $\partial x / \partial \bar{x}$ is the Jacobian matrix of the coordinate transform with componentns $\partial x_i / \partial \bar{x}_j$. In other words, the metric tensor transforms according to

$$\bar{G} = \frac{\partial x}{\partial \bar{x}}^T G \frac{\partial x}{\partial \bar{x}}. \tag{10}$$

Under composition of several coordinate transformations, $\tilde{x} = \tilde{x}(\bar{x}) = \tilde{x}(\bar{x}(x))$, the process is nested,

$$ds^2 = d\tilde{x}^T \frac{\partial \bar{x}}{\partial \tilde{x}}^T \frac{\partial x}{\partial \bar{x}}^T \mathrm{G} \frac{\partial x}{\partial \bar{x}} \frac{\partial \bar{x}}{\partial \tilde{x}} d\tilde{x}, \tag{11}$$

$$\tilde{\mathrm{G}} = \frac{\partial \bar{x}}{\partial \tilde{x}}^T \frac{\partial x}{\partial \bar{x}}^T \mathrm{G} \frac{\partial x}{\partial \bar{x}} \frac{\partial \bar{x}}{\partial \tilde{x}}, \tag{12}$$

which can also be seen directly from the chain rule for the Jacobian matrices,

$$\frac{\partial x}{\partial \tilde{x}} = \frac{\partial x}{\partial \bar{x}} \frac{\partial \bar{x}}{\partial \tilde{x}}. \tag{13}$$

All the points with unit distance from a given central point—a unit ball—constitute an ellipsoid

$$\Delta x^T \mathrm{G} \Delta x = 1. \tag{14}$$

The cross section of this ellipsoid with a principal plane in a given coordinate is obtained by setting the corresponding $\Delta x_i = 0$, reducing the ellipsoid to an ellipsis (*Pant & Farup, 2012*),

$$\begin{pmatrix} \Delta x_1 & \Delta x_2 \end{pmatrix} \begin{pmatrix} g_{11} & g_{12} \\ g_{21} & g_{22} \end{pmatrix} \begin{pmatrix} \Delta x_1 \\ \Delta x_2 \end{pmatrix} = 1, \tag{15}$$

with angle $\theta$ and semi-axes $a$ and $b$ given by

$$\tan(2\theta) = \frac{2g_{12}}{g_{11} - g_{22}}, \tag{16}$$

$$a = \frac{1}{\sqrt{g_{22} + g_{12} \cot\theta}}, \tag{17}$$

$$b = \frac{1}{\sqrt{g_{11} - g_{12} \cot\theta}}. \tag{18}$$

## SYSTEM ARCHITECTURE

Since the computation of generic colour space transforms and, in partiular the composition of their Jacobian matrices can be a tedious and error-prone process (see, e.g., *Pant & Farup, 2012*), an object-oriented framework for transforming colour and metric data between colour spaces has been implemented as a Python package `colour`. The package consists of six partially interdependent modules `space`, `data`, `metric`, `tensor`, `statistics` and `misc`. The relationship between the modules is shown in Fig. 1. In the figure, the arrows indicate dependencies between the modules in the form of Python imports. Each of the modules contain functions, classes and predefined objects with the purpose of simplifying the implementation of new colour spaces and metrics.

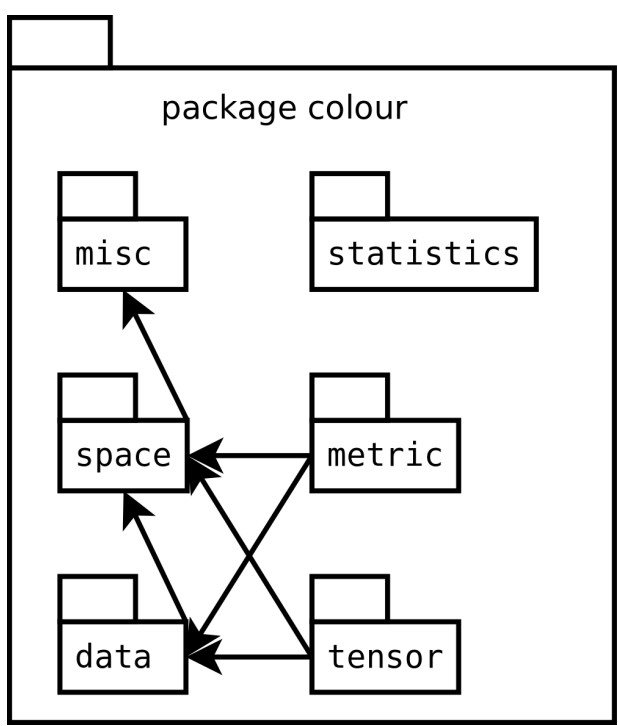

**Figure 1** **Structure of the modules within the `colour` package.** The arrows indicate dependencies in the form of Python imports.

## Representing colour spaces

The core functionality of the colour space and colour metric transforms is found in the space module. The basic idea in designing the object oriented framework is to realise a colour space as an object, and to facilitate the construction of new such objects by providing classes for transforming new colour spaces from already existing ones. The class hierarchy which constitutes the core of the of the module, is shown in Fig. 2. All boxes represent classes, and the arrows denote class inheritance. Italicized method names indicate methods that should be overridden in a subclass. Details about attributes and auxiliary methods etc. have been left out for readability.

All colour space objects must derive from the abstract Space class, and as such implement the methods `to_XYZ` and `from_XYZ` for converting colour data between the XYZ colour space and the colour space represented by the object, and the methods `jacobian_XYZ` and `inv_jacobian_XYZ` for computing the corresponding Jacobian matrix and its inverse. The two latter methods are implemented in Space as inverses of each other, so the subclasses only need to implement one of them—the other one can be inherited. The colour data is represented as $N \times 3$ NumPy (*Oliphant, 2007*) ndarrays, and the Jacobian matrices as $N \times 3 \times 3$ ndarrays.

All transformations between colour spaces must go through XYZ, which thus serves a special role, and has a separate class of its own. Here, the transformations are simply the identity transform, and the Jacobian matrices are identity matrices. All other colour spaces

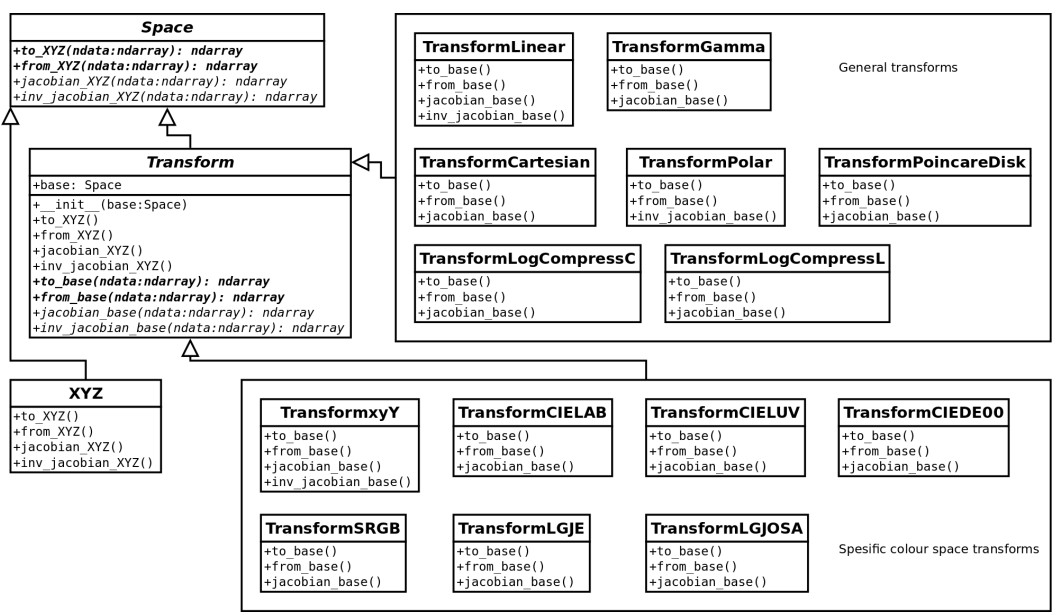

**Figure 2** **Structure of the classes within the space module.** Data types and parameters are not shown for the derived methods.

will be built by transforming colour data from an already existing colour space, starting from XYZ. To facilitate making the specific transforms, an abstract class `Transform` is provided. During instantiation of a transformed colour space object, the base space of the transformation has to be set. The virtual methods `to_base` and `from_base` for converting colour data to and from the base base, and `jacobian_base` and `inv_jacobian_base` for computing the Jacobian matrix and its inverse between the current space and the base space must be provided in the derived classes. The methods `to_XYZ`, `from_XYZ`, `jacobian_XYZ`, and `inv_jacobian_XYZ` are implemented in the base class `Transform` using the transformation between the current space and the base space (provided by derived classes) and the corresponding transformations in the base class, see Eq. (12). Hence, there is no need to reimplement these in the derived classes. Finally, the concrete colour space tranforms are implemented as classes `TransformXXX` derived from `Transform`. They must all implement the methods `to_base`, `from_base` and either `jacobian_base` or `inv_jacobian_base`. The remaining methods will be inferred by inheritance. In some cases, though, it is more efficient to provide more methods in order to reduce the computational cost. For example, in `TransformLinear`, both methods `jacobian_base` and `inv_jacobian_base` are provided in order to avoid inverting every single Jacobian matrix for large data sets.

## Representing colour and metric data

The colour space objects constructed by the method described above, will convert colour data represented as $N \times 3$ ndarrays ($N$ colour data points). For real-life applications, colours some times come as single data points (3-vectors), some times as lists of colour data ($N \times 3$ matrices), and some times as images ($M \times N \times 3$ arrays). In order for the

| Data |
|---|
| +data: dict |
| +\_\_init\_\_(space:Space,data:ndarray) |
| +set(space:Space,data:ndarray) |
| +get(space:Space): ndarray |
| +get_linear(space:Space): ndarray |
| +new_white_point(space:Space,from_white:Data, to_white:Data): Data |

| TensorData |
|---|
| +points: Data |
| +metrics: dict |
| +\_\_init\_\_(space:Space,points_data:Data,metric_data:ndarray) |
| +set(space:Space,points_data:Data,metric_data:ndarray) |
| +get(space:Space): ndarray |
| +get_ellipse_parameters(space:Space,plane:slice, scale:float=1): ndarray |
| +get_ellipses(space:Space,plane:slice,scale:float=1): list |

**Figure 3** The `Data` and `TensorData` classes for keeping track of colour data and metric data, respectively.

user not having to deal with converting back and forth between these formats, as well as remembering in which colour space all the data is given, a separate class `Data` for storing colour data has been implemented as part of the `data` module, cf. Fig. 3. Again, the boxes denote classes. In this module, there are no inheritance relationship between the classes, but they are related by the `TensorData` class having an attribute of the `Data` type.

A colour `Data` object can be instantiated with single colour data, lists of colour data, or colour images in any implemented colour space. The `Data` object takes the colour space (object) of the data as an argument, and keeps a dictionary of the colour spaces in which the data in question has been computed. When the colour data in a given space is requested (using the `get` method), it first checks the dictionary whether it has already been computed. If not, it is computed, stored in the dictionary and returned. All the actual computations are taken care of by the hierarcy of colour space objects representing the transforms necessary for building the colour space.

A similar approach is taken for colour metric data in the form of colour tensors, see Fig. 3. In this case, both the locations of the colour metrics (as colour `Data`), and the metrics themselves are represented in the class. Like for the colour data, a dictionary of the computed tensors is maintained. For the conversion between the different colour spaces, the Jacobian matrices are applied according to Eq. (10). The nested tensor transforms, Eq. (12), are implicitly taken care of by the colour space class hierarchy without the user having to interfere.

## Colour metrics, tensors and statistics

The four remaining modules, `metric`, `tensor`, `statistics`, and `misc` contain separate functions (not part of the class hierarchy) for computing various properties of colour data, colour transforms, and sets of these. The `metric` module has functions for computing the most common colour metrics, such as the standard CIE $\Delta E_{ab}$ and $\Delta E_{uv}$ metrics, CIEDE2000 (*Luo, Cui & Rigg, 2001*), the different versions of the DIN99 metric as described by *Cui et al. (2002)*, the log-compressed OSA-UCS metric $\Delta E_E$ by *Oleari, Melgosa & Huertas (2009)*, as well as a general Euclidean distance and the Poincare disk metric developed in Reference (*Farup, 2014*) in any colour space. All these functions take two colour data objects of the same size as arguments, and return an $N$-vector of colour differences.

The `tensor` module has functions for computing the metric tensors corresponding to the metrics in the `metric` package. The functions take one colour `Data` object as argument, and

returns the corresponding `TensorData` object. The `statistics` module contains functions for calculating various statistics of colour metric data, such as the STRESS measure (*García et al., 2007*) and Pant's R-values (*Pant & Farup, 2012*; *Pant, Farup & Melgosa, 2013*). The `misc` module contains miscellaneous supporting functions.

### Computational complexity

The framework has been implemented using NumPy (*Oliphant, 2007*) `ndarrays`, and all operations for colour and metric conversion are vectorised. Thus, all the real computations take place using the highly optimised underlying libraries for matrices, such as LAPACK (*Anderson et al., 1999*) etc. No loops over individual colour data are implemented in the high-level language. Thus, there are only two sources of computational overhead by using the framework.

First, there are the function calls associated with computing the transformations between a given colour space and its bases all the way back to the CIEXYZ colour space. These function calls will only take place once per transformation call. This can be a significant overhead when the data set consists of only one or very few data points, but then the computation is very quick anyway. For bigger colour data sets, such as images, this will represent only very few function calls (given by the number of steps in the transformation from CIEXYZ to the given space), and thus be negligible in comparison with the real computation, which takes place at the highly optimised low-level code.

Secondly, all colour and metric conversions go through the CIEXYZ colour space. When converting between two colour spaces based on a common basis *other than* CIEXYZ, an unneccesary conversion back and forth between this common basis and CIEXYZ will inevitably take place. It would, in principle, be possible to eliminate this by advanced optimisation techniques, but since the computations are already fast (fractions of a second even for quite large images), and the goal of the framework has been ease of implementation rather than computational efficiency, this has not been prioritized.

Not all of the operations in the `statistics` module are vectorised, although this would in principle also be possible. The reason for this is that they are mainly meant for colour research applications, and as such, they are not expected to be used in production. For the relevant use in research, data collection etc. will be much more time consuming than the actual computations, so computational efficiency has not been emphasized in this part of the framework.

## EXAMPLE APPLICATION

In order to demonstrate the power of the proposed approach, a simple demo application is shown in Fig. 4. In this short code (less than a page), (i) a new colour space is implemented, (ii) individual colours, lists of colours and a colour image is converted to the new colour space, (iii) the tensorial data corresponding to the MacAdam ellipses is converted by the help of the Jacobian matrices of the transformation to the new space, and (iv) the colour difference between two images is computed as a Eucildean distance in the newly constructed colour space. These operations would normally require days of programming, but with the use of the proposed framework it is all achieved by a few lines of code.

```
1   import numpy as np
2   import matplotlib.pyplot as plt
3   from colour.space import TransformLinear, TransformGamma, xyz, cielab, srgb
4   from colour.data import Data, g_MacAdam
5   from colour.metric import dE_ab, dE_00, euclidean
6   from colour.misc import plot_ellipses
7
8   # Construct the IPT colour space
9   A = np.array([[.4002, .7075, -.0807],      # parameters for the transform
10                 [-.228, 1.15, .0612],        # from XYZ to IPT
11                 [0, 0, .9184]])              #
12  B = np.array([[.4, .4, .2],                 #
13                 [4.455, -4.850, .3960],      #
14                 [.8056, .3572, -1.1628]])    #
15  gamma = 0.43                                #
16  ipt = TransformLinear(TransformGamma(TransformLinear(xyz, A), .43), B)
17
18  # Convert colours in different formats
19  white = Data(cielab, [100, 0, 0])           # single colour
20  print(white.get(ipt))
21
22  grays = Data(cielab, [[100, 0, 0], [50, 0, 0], [0, 0, 0]])
23  print(grays.get(ipt))                       # list of colours
24
25  im = Data(srgb, plt.imread('camera.png')) # colour image
26  im_ipt = im.get(ipt)
27  plt.imsave('camera_i.png', im_ipt[:,:,0], vmin=0, vmax=1, cmap=plt.cm.gray)
28  plt.imsave('camera_p.png', im_ipt[:,:,1], vmin=-1, vmax=1, cmap=plt.cm.gray)
29  plt.imsave('camera_t.png', im_ipt[:,:,2], vmin=-1, vmax=1, cmap=plt.cm.gray)
30
31  # Load the MacAdam ellipses and show them in the PT plane
32  mca = g_MacAdam()
33  mca_points_ipt = mca.points.get(ipt)
34  plt.plot(mca_points_ipt[:,1], mca_points_ipt[:,2],'.')
35  plot_ellipses(mca.get_ellipses(ipt, mca.plane_12, 10))
36  plt.axis('equal')
37  plt.savefig('mca_ellipses_ipt.pdf')
38
39  # Compute colour difference of images
40  im_clip = Data(srgb, plt.imread('camera_clip.png'))
41  plt.imsave('dE_ab.png', dE_ab(im, im_clip), cmap=plt.cm.gray)
42  plt.imsave('dE_00.png', dE_00(im, im_clip), cmap=plt.cm.gray)
43  plt.imsave('dE_ipt.png', euclidean(ipt, im, im_clip), cmap=plt.cm.gray)
```

**Figure 4**  Small demo of the colour package.

In lines 3–6, the library is imported. In a real application, one would normally only import the `colour` package (`import colour`), and refer to the elements as, e.g., `colour.space.TransformLinear` etc., but here the specific classes, objects and functions needed are imported specifically, simply in order to reduce the size and improve the readability of the remaining code.

The transformation to the IPT colour space (*Ebner & Fairchild, 1998*) is composed of a linear transform from XYZ followed by a gamma correction, followed by final linear

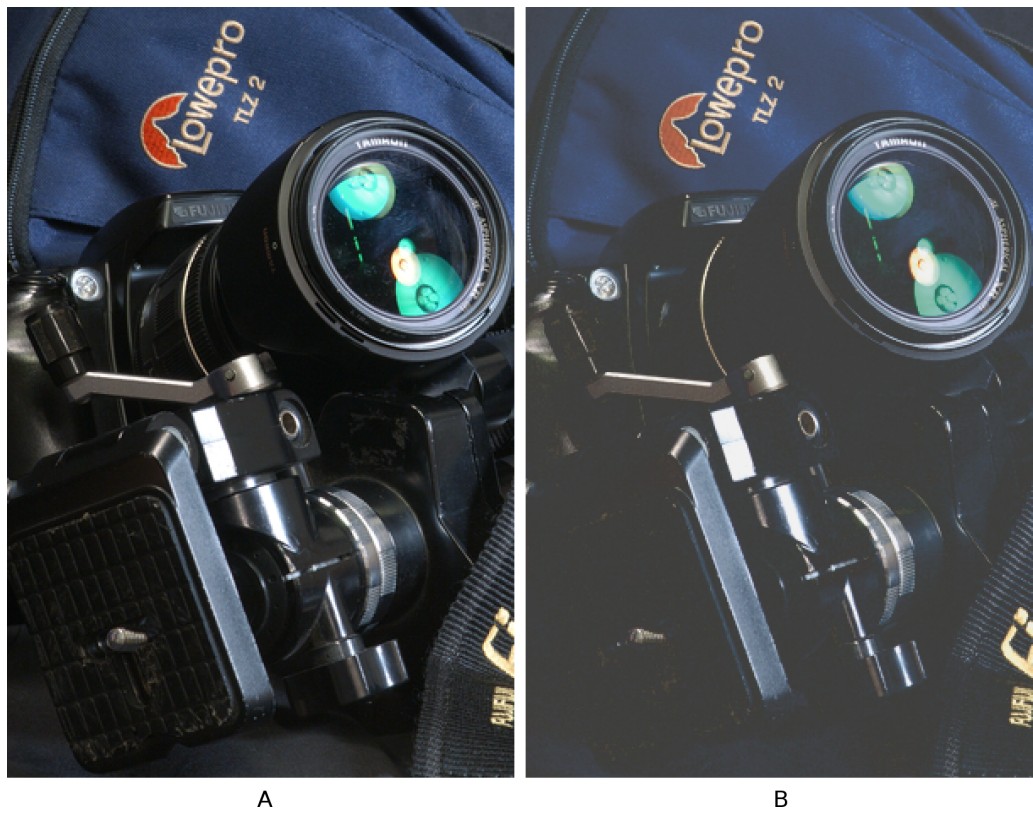

**Figure 5** An image (A) and its gamut clipped version (B).

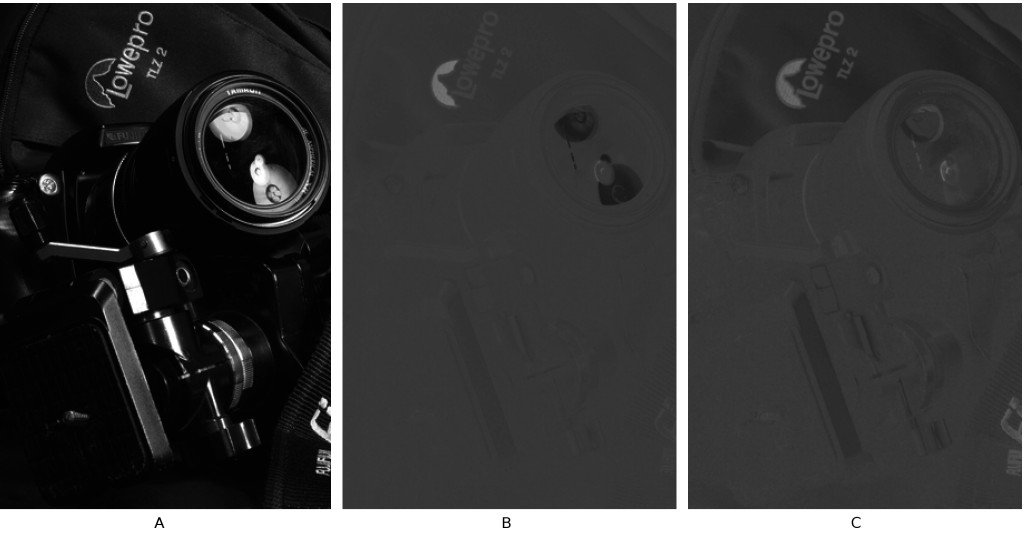

**Figure 6** The *I* (A), *P* (B), and *T* (C) planes of the image shown in Fig. 5A.

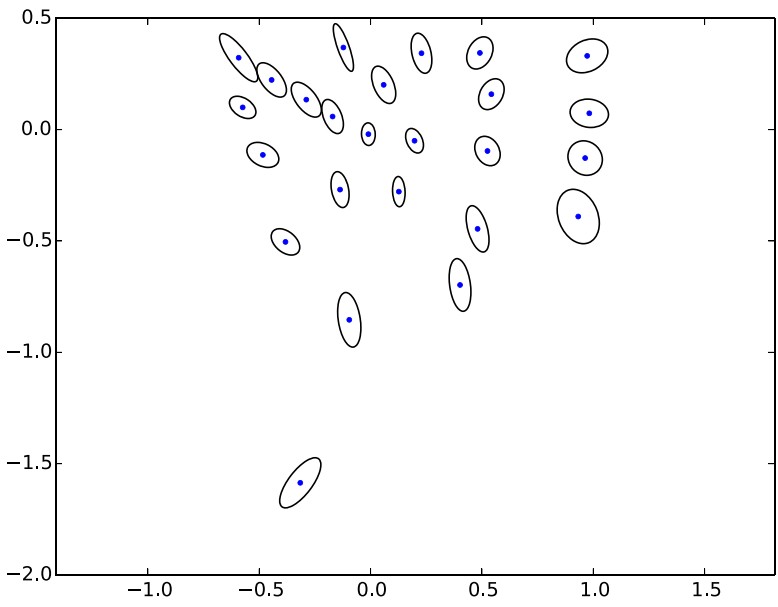

**Figure 7** **The MacAdam ellipses plotted in the PT-plane of the IPT colour space.**

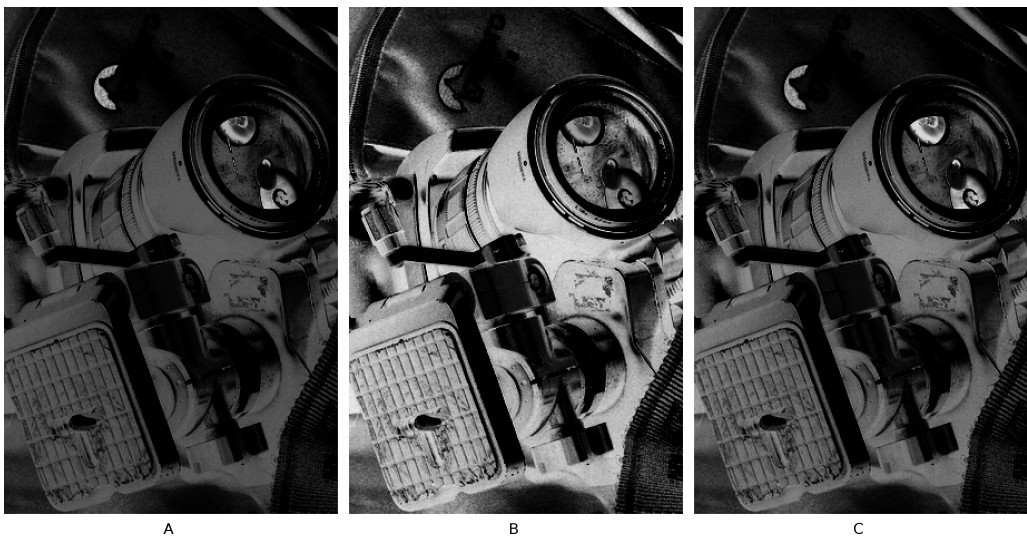

**Figure 8** **Difference maps of the two images shown in Fig. 5 for $\Delta E_{ab}$ (A), $\Delta E_{00}$ (B), and the Euclidean distance in the IPT colour space (C).**

transform. The code for constructing this colour space is given in lines 9–16 of Fig. 4. It should be noted that the programmer does not need to specify anything about the computation of the corresponding Jacobian matrices—everyting is taken care of by the constructors of the `Transformation` classes.

Once the colour space is constructed, the `Data` class can use it for converting colours in various formats such as single data points—lines 19–20—giving

`[9.99987871e-01 1.16264986e-03 1.69020684e-06]`,

lists of colour points—lines 22–23—giving

```
[[9.99987871e-01 1.16264986e-03 1.69020684e-06]
 [4.83120653e-01 5.61706973e-04 8.16583738e-07]
 [0.00000000e+00 0.00000000e+00 0.00000000e+00]],
```

and even colour images (lines 25–26). The individual IPT colour planes of the image shown in Fig. 5A resulting from this (lines 27–29) are shown in Fig. 6.

In order to demonstrate also the transformation of tensorial colour data, the code in lines 32–37 loads, transforms, and plots the MacAdam ellipses (*MacAdam, 1942*) in the PT-plane of the IPT space. The latter includes the tedious process of computing the transformation of the corresponding metric tensors according to Eq. (10). The resulting plot is shown in Fig. 7.

Similarly, the `colour.metric` module can compute colour differences of colour data in any format, including images. For example, the code in lines 41–43 of Fig. 4 computes the difference maps of the two images shown in Fig. 5 for $\Delta E_{ab}$, $\Delta E_{00}$ and the Euclidean distance in the newly implemented IPT colour space. The results are shown in Fig. 8.

Please note that the *entire* code used to generate Figs. 5–8 is shown in Fig. 4.

## CONCLUSION

An object-oriented computational framework for colour metrics and colour has been designed and implemented in Python. The framework strongly simplifies the implementation of new colour spaces for transfroming colour data and as well as tensorial colour metric data between the various colour spaces without compromising too much on the computational complexity. The code is freely available at GitHub (https://github.com/ifarup/colourspace). Future extensions could include ICC support, computation geodesics based on the colour metrics (*Pant & Farup, 2013*), computation and representation of colour gamuts (*Bakke, Farup & Hardeberg, 2010*), as well as gamut mapping algorithms (*Alsam & Farup, 2009*) in any colour space, and under any colour metric.

### Funding

This research has been supported by the Research Council of Norway through project no. 221073 'HyPerCept – Colour and quality in higher dimensions'. The funders had no role in study design, data collection and analysis, decision to publish, or preparation of the manuscript.

### Grant Disclosures

The following grant information was disclosed by the author:
Research Council of Norway: 221073.

## PeerJ Computer Science

## Competing Interests

The author declares there are no competing interests.

## Author Contributions

- Ivar Farup conceived and designed the experiments, performed the experiments, analyzed the data, contributed reagents/materials/analysis tools, wrote the paper, prepared figures and/or tables, performed the computation work, reviewed drafts of the paper.

## Data Availability

https://github.com/ifarup/colourspace.

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
