# Peer review of "A computational framework for colour metrics and colour space transforms"

_PeerJ Computer Science, doi:10.7717/peerj-cs.48_

## Round 0.1 · original submission · Major Revisions

Dear author,

All in all the referees would like the paper to be much clearer. The innovation spelled out and the Figures thoroughly explained. Below is a summary of their remarks.

The author presents an interesting implementation related to color metrics and color space transformation. It is not clear whether there are other such implementations, and if yes – one should see comparisons to them (see metrics, for instance, in: M. M. Deza and E. Deza, Encyclopedia of Distances, Springer-Verlag Berlin Heidelberg 2009). The innovations in the paper should be emphasized.

The word "CIEDE2000" is used as both of metric and space. Readers may be confused. CIEDE2000 is usually used as a color difference metric. The author should define the CIEDE2000 color space.
The author claims that new color spaces can be created. However, there is no discussion about the meaningful created color spaces.

The author should explain better figures 5 to 8.
More figures in Sections 2 and 3 will help to illustrate the work applications.

More examples are needed in the "Example Application" Section 4.

The Conclusion (Section 5) should be expanded.

Reviewer 1 ·

Basic reporting

The author presents an interesting material related to colour metrics and colour space transformation . It will be nice if he explains better the figures 5 to 8 , and organizes better the references.

Experimental design

Nice work . However, the inclusion of more figures in section 2 and 3 will help to illustrate the work applications, as well as more examples in the "Example Application" section (4).

Validity of the findings

The Conclusion (section 5) deserve be expanted.

Additional comments

How about some comparison with other usual metrics for instance those presented in: M. M. Deza and E. Deza, Encyclopedia of Distances, Springer-Verlag Berlin Heidelberg 2009

Reviewer 2 ·

Basic reporting

The word "CIEDE2000" is used as both of metric and space. Readers may confuse. CIEDE2000 is usually used as a color difference metric. You should clearly define the CIEDE2000 color space.

Experimental design

In the manuscript, it is written that new color spaces can be created. However, there is no discussion about the meaningful color spaces. It is important to derive new color spaces from the derived framework.

Validity of the findings

The discussion of computational complexity or processing speed lacks.

Reviewer 3 ·

Basic reporting

It is unclear whether others have already implemented a similar job and a comparison between the proposed method and the other.

Experimental design

no comment

Validity of the findings

There is uncertainty regarding the strengths of the work. it is necessary to explain the innovative points.
It is not clear the robustness of the method.
there are no news from the point of view of research but only from the implementation point of view.

---

## Round 0.2 · accepted · Accept

The corrections and clarifications are satisfactory